

# *AlleleShift:* an R package to predict and visualize population-level changes in allele frequencies in response to climate change

Roeland Kindt

World Agroforestry, CIFOR-ICRAF, Nairobi, Kenya

Corresponding author
Roeland Kindt, R.Kindt@CGIAR.org

## ABSTRACT

**Background**. At any particular location, frequencies of alleles that are associated with adaptive traits are expected to change in future climates through local adaption and migration, including assisted migration (human-implemented when climate change is more rapid than natural migration rates). Making the assumption that the baseline frequencies of alleles across environmental gradients can act as a predictor of patterns in changed climates (typically future but possibly paleo-climates), a methodology is provided by *AlleleShift* of predicting changes in allele frequencies at the population level.

**Methods**. The prediction procedure involves a first calibration and prediction step through redundancy analysis (RDA), and a second calibration and prediction step through a generalized additive model (GAM) with a binomial family. As such, the procedure is fundamentally different to an alternative approach recently proposed to predict changes in allele frequencies from canonical correspondence analysis (CCA). The RDA step is based on the Euclidean distance that is also the typical distance used in Analysis of Molecular Variance (AMOVA). Because the RDA step or CCA approach sometimes predict negative allele frequencies, the GAM step ensures that allele frequencies are in the range of 0 to 1.

**Results**. *AlleleShift* provides data sets with predicted frequencies and several visualization methods to depict the predicted shifts in allele frequencies from baseline to changed climates. These visualizations include 'dot plot' graphics (function *shift.dot.ggplot*), pie diagrams (*shift.pie.ggplot*), moon diagrams (*shift.moon.ggplot*), 'waffle' diagrams (*shift.waffle.ggplot*) and smoothed surface diagrams of allele frequencies of baseline or future patterns in geographical space (*shift.surf.ggplot*). As these visualizations were generated through the *ggplot2* package, methods of generating animations for a climate change time series are straightforward, as shown in the documentation of *AlleleShift* and in the supplemental videos.

**Availability**. *AlleleShift* is available as an open-source R package from https://cran.r-project.org/package=AlleleShift and https://github.com/RoelandKindt/AlleleShift. Genetic input data is expected to be in the *adegenet::genpop* format, which can be generated from the *adegenet::genind* format. Climate data is available from various resources such as *WorldClim* and *Envirem*.

**Subjects** Bioinformatics, Genetics, Genomics, Climate Change Biology

---

**Keywords** Climate change, Genomic adaptation, Ecological modelling, R package, Redundancy Analysis, Genome Wide Association Studies, Environmental Association Analysis, BiodiversityR package, Analysis of Molecular Variance, Euclidean distance

# INTRODUCTION

There is clear evidence of anthropogenically induced climate change, with our planet facing a climate emergency (*Ripple et al., 2020*). Anticipating climate change, many countries are developing National Adaptation Plans (NAPs; https://www4.unfccc.int/sites/napc). Specifically for forests and trees, technical guidelines have recently been prepared on the integration of forests, agroforestry and trees in the formulation and implementations of NAPs (*Meybeck et al., 2020*). *Stanturf et al. (2015)* provide a practical framework and a stoplight tool to plan for climate change mitigation and adaptation in forest and landscape restoration initiatives.

For loci involved in adaptation, shifts in allele frequencies (and changes in phenotypes as a result) can be anticipated (*Günther & Coop, 2013*; *Stange, Barrett & Hendry, 2020*). Although the methods involved are far from straightforward, statistical approaches such as Genome Wide Association Studies (GWAS) and Environmental Association Analyses (EAA) can be applied to genomic data to postulate genes and specific alleles involved in climate change responses (*Luikart et al., 2018*; *Anderson & Song, 2020*; *Waldvogel et al., 2020*). *AlleleShift* predicts shifts in allele frequencies for those loci predicted by GWAS or EAA to be associated with adaptive traits.

The methodology used in *AlleleShift* exploits the analogy between analysing a community matrix (consisting of sites as rows, species as columns and abundances as cell values) and a genetic matrix (consisting of populations or individuals as rows, alleles as columns and allele counts as cell values). As a consequence, the common constrained ordination method in the field of community ecology of redundancy analysis (RDA; *Legendre & Legendre, 2012*, pp. 629–661) can be applied.

RDA is based on Euclidean distances and without explanatory variables is equivalent to principal components analysis (PCA). Various recent studies of adaptative genetic variation have also used the RDA methodology (e.g., *Razgour et al., 2019*; *Capblancq et al., 2020*; *Nelson, Motamayor & Cornejo, 2020*; *Temunović et al., 2020*). Analysis of a genetic matrix via Euclidean distances is appropriate for several reasons:

- Euclidean distances are also used in Analysis of Molecular Variance (AMOVA; (*Excoffier, Smouse & Quattro, 1992*; *Meirmans & Liu, 2018*; *Michalakis & Excoffier, 1996*)) and it can be demonstrated (see examples for the `AlleleShift::amova.rda` function and possibly also compare with AMOVA analysis in GenAlEx; (*Peakall & Smouse, 2012*)) that RDA provides the same information on squared Euclidean distances and mean squares as an AMOVA analysis.
- In Article S1, I demonstrate how Euclidean distances between `adegenet::genpop` objects are linearly related to the Euclidean distances between the centroids obtained from a PCA analysis of `adegenet::genind` objects. As a corollary, shifts of populations can be understood as the average shift of individuals in ordination space.
- Recently, I also showed (*Kindt, 2020a*) how RDA can be directly interpreted in terms of Sums-of-Squares of AMOVA by analysing distances from individuals to centroids and among centroids.

The RDA model is calibrated with an explanatory data set that documents the environmental values of populations in the baseline climate. The prediction function uses the values of populations in the changed (future or past) climate as explanatory variables to predict changes in allele counts and frequencies in the changed climate. As possibly negative allele counts can be predicted by the calibrated RDA, as with the example data set shown below, a second calibration step is implemented that guarantees that allele frequencies are within the biologically realistic interval between 0 and 1.

By using RDA and avoiding negative allele frequencies, *AlleleShift* is fundamentally different from the protocol developed by *Blumstein et al. (2020)* based on canonical correspondence analysis (CCA; *Legendre & Legendre, 2012*, pp. 661–673). Besides the advantage of RDA to reflect Euclidean distances, a disadvantage of CCA is that the interpretation of species scores in ordination diagrams is more complex in showing the peak of its unimodal distribution against a vector of an explanatory variable (see Fig. 3 in *Ter Braak (1987)*; Fig. 11.9 in *Legendre & Legendr (2012)*; or Figure 10.13 in *Kindt and Coe, 2005*).

## MATERIALS & METHODS

### Data import

Genetic response data (including a matrix with populations as rows and allele counts as columns) for the calibration of the `AlleleShift::count.model` and prediction via `AlleleShift::count.pred` is required to be in the `adegenet::genpop` format. Individual-based data that are in the `adegenet::genind` format can be converted into the genpop format via the `adegenet::genind2genpop` function. The *adegenet* and *poppr* packages provide various methods of importing data from other software application formats into the *genind* format, such as `adegenet::import2genind` and `poppr::read.genalex`. Environmental data of populations, used as explanatory variables in redundancy analysis (RDA), is expected to be provided as a data.frame with the same sequence of populations as the genetic response data (this is a general requirement for community ecology methods in the *vegan* and *BiodiversityR* packages; a check is available via `BiodiversityR::check.datasets`). Whereas environmental data typically is baseline and changed (bio)climatic data such as is available from WorldClim (*Fick & Hijmans, 2017*), ENVIREM (*Title & Bemmels, 2018*), CHELSA (*Karger, Schmatz & Dettling, 2020*) or PaleoClim (*Brown et al., 2018*), it is also possible to expand input to other data available for species distribution modelling (a recent overview of available data sets is provided by *Booth (2018)*.

### Data analysis

Prior to calibrating the models that predict allele frequencies from bioclimatic explanatory data, it is recommended to reduce the explanatory variables to a subset where the Variance

Inflation Factor (VIF) is below a predefined threshold for each variable. Such methods are also recommended for regression analysis (*Fox & Monette, 1992*) and species distribution modelling (*Kindt, 2018*). With this approach, it is easy to select the same variables from future data sets and comparison with other studies may also become easier. VIF analysis, and an additional feature of forcing the algorithm to keep preselected variables within the final subset, is available via `AlleleShift::VIF.subset` (Table 1). This is a function that uses `BiodiversityR::ensemble.VIF.dataframe` internally after a first step of removing all explanatory variables that have correlations larger than the VIF threshold with the preselected variables. There is an option to generate a correlation matrix for the final subset of variables (Fig. 1). I also recommend conducting the VIF analysis for the genetic response data (see discussion and Fig. 1). I further advise to remove any individuals with partially missing genetic or (bio)climatic data prior to the analysis. As a sensitivity analysis, results could possibly be compared with models calibrated with data where missing genotypes were replaced by genotypes predicted by the impute function of the *LEA* package (*Gain & François, 2021*).

Prior to model calibration, I suggest checking (according to criteria described below) for the shifts of populations in environmental space between the baseline and changed climates. This can be achieved via function `AlleleShift::population.shift`, which draws arrows between each population in the baseline and changed climate. There are alternatives of using principal components analysis (PCA) or redundancy analysis (RDA) to generate the ordination diagrams (Fig. 2). What is important to check in the ordination graphs is whether populations shift in a similar fashion, as that will facilitate the interpretation of predicted shifts in allele frequencies. If some populations would show a different trend in the ordination graph, their allele frequencies would also be expected to change in a different way.

With the selected genetic and explanatory data, model calibration can be done. In a first step, a RDA model is fitted (`AlleleShift::count.model`) that can predict counts of alleles in baseline or changed climates (`AlleleShift::count.pred`). The user has the option also to obtain results for the canonical correspondence analysis procedure of *Blumstein et al. (2020)* with the count model via argument `cca.model=TRUE`. In the second step, the predicted allele counts for the baseline climate serve as explanatory variables for the calibration of a generalized linear model (GAM via `mgcv::gam`; *Wood, 2004*) with the baseline allele frequencies as response and a binomial family function (`AlleleShift::freq.model`). This procedure ensures that predictions are within the realistic interval for frequencies between 0 to 1. Function `AlleleShift::freq.pred` allows the prediction of allele frequencies for baseline and changed climates. A simulation study (Article S2, using a subset of a simulation study available from *Frichot & François (2015)*) and the results below show that the predicted frequencies approximate the expected frequencies well.

The output of the two steps (RDA followed by GAM) is a data.frame as shown in Table 2 (to fit printing space available here, the data is shown in a transposed format where rows and columns were swapped). All figures shown in this manuscript were obtained with the example data sets of `AlleleShift::Poptri.genind` (individually-based allele

**Table 1  Functions found in *AlleleShift* and their short descriptions.**

| Function | Description |
| --- | --- |
| Preparation | |
| VIF.subset | Reduce the number of explanatory variables through Variance Inflation Factor analysis, with an option to plot a correlation matrix (GGally::ggcorr). Internally, the function calls the BiodiversityR::ensemble.VIF.dataframe function. |
| environmental.novel | Identify populations with some values of the explanatory variables that are outside the range of values used for calibration. |
| Analysis | |
| count.model | Calibration of RDA model (vegan::rda) with baseline allele counts as response and baseline bioclimatic variables as explanatory variables |
| count.pred | Prediction of allele counts (vegan::predict.rda) from explanatory variables. Explanatory variables correspond to the baseline climate to check the calibration |
| freq.model | Calibration of GAM model (mgcv::gam) with baseline allele frequencies as response and predicted baseline counts from the RDA model as explanatory variables |
| freq.pred | Prediction of allele frequencies (mgcv::predict.gam) from the predicted alleles counts of count.pred |
| amova.rda | Perform AMOVA with the outputs from RDA. The function returns an output that is similar to the output of poppr::poppr.amova so that results can be readily compared |
| Visualization | |
| population.shift | Shifts of populations in environmental space, with superellipses (ggforce::geom_mark_ellipse) and arrows between baseline and changed positions to show climatic shifts. Internally, the function calls vegan::ordiplot and various helper functions from BiodiversityR (*Kindt, 2020a*) are used |
| freq.ggplot | Plots of baseline allele frequencies against predicted allele frequencies. Data points can be coloured differently by population or by allele |
| shift.dot.ggplot | Shifts of Allele Frequencies as Response to Climate Change |
| shift.pie.ggplot | Shifts of Allele Frequencies as Response to Climate Change via ggforce::geom_arc_bar |
| pie.baker | Helper function to prepare data for shift.pie.ggplot from the output of freq.pred. |
| shift.moon.ggplot | Shifts of Allele Frequencies as Response to Climate Change via gggibbous::geom_moon |
| moon.waxer | Helper function to prepare data for shift.moon.ggplot from the output of freq.pred |
| shift.waffle.ggplot | Shifts of Allele Frequencies as Response to Climate Change. Graphics are similar to waffle::waffle, but the graph is made *de novo* in *AlleleShift* |
| waffle.baker | Helper function to prepare data for shift.waffle.ggplot from the output of freq.pred |
| shift.surf.ggplot | Shifts of Allele Frequencies as Response to Climate Change, plotted in geographical space through smoothed regression surfaces (vegan::ordisurf) |

**Notes.**

BiodiversityR (*Kindt & Coe, 2005*; version 2.12-3 used in this manuscript); *ggforce* (*Pedersen & Robinson, 2020*; version 0.3.2); *gggibbous* (*Bramson, 2019*; version 0.1.0); mgcv (*Wood, 2004*); version 1.8-31); *poppr* *Kamvar, Tabima & Grünwald, 2014*; version 2.8.6); vegan (*Oksanen et al., 2020*; version 2.5-6).

A

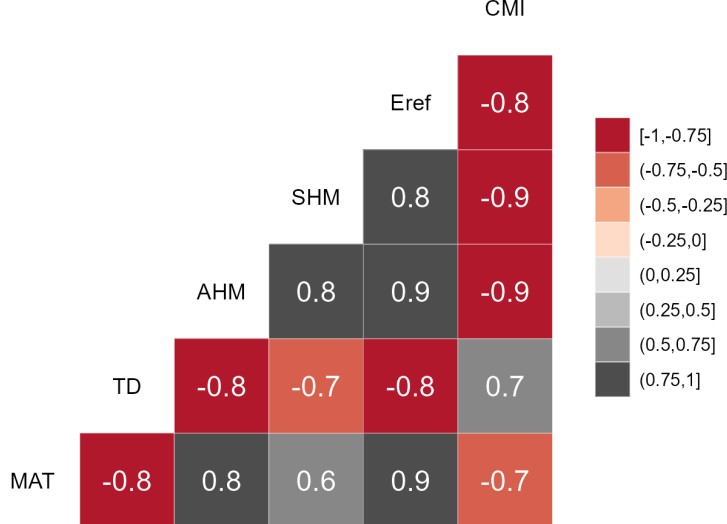

B

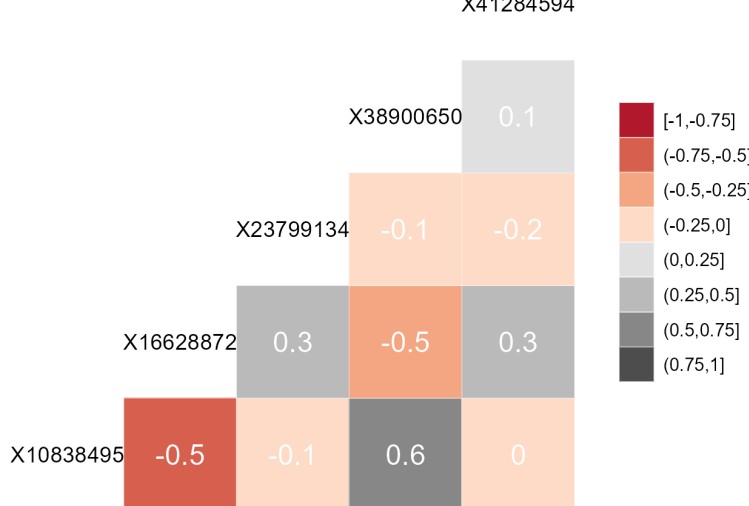

**Figure 1 Correlation matrices.** (A) Correlation matrix plot for the final subset of bioclimatic variables selected by the VIF.subset function. CMI, climatic moisture index; Eref, reference atmospheric evaporative demand; SHM, summer heat-to-moisture index; AHM, annual heat-to-moisture index; TD, continentality; MAT, mean annual temperature. (B) Correlation matrix plot for the minor alleles.

counts), `AlleleShift::Poptri.baseline.env` (climatic descriptors of the populations in the baseline climate), `AlleleShift::Poptri.future.env` (climatic descriptors of the populations in the future climate) and `AlleleShift::Poptri.loc` (geographical coordinates of the populations). These data sets were converted from the example data sets provided by *Blumstein et al. (2020)* for *Populus trichocarpa*. It can be seen for population

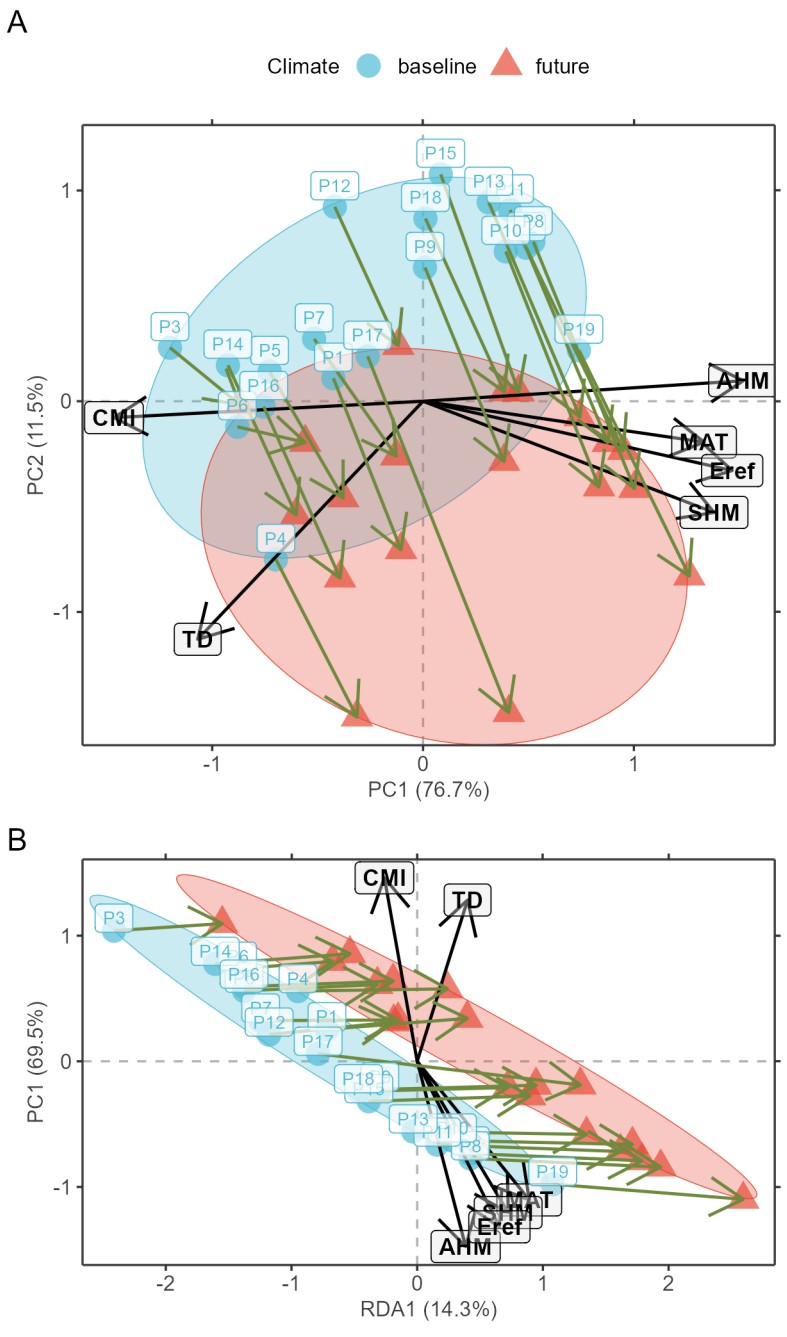

**Figure 2 Ordination graphs.** (A) Principal Component Analysis (PCA) ordination graph for shifts in populations in environmental space. (B) Redundancy Analysis (RDA) ordination graph for shifts in populations with climate (a categorical variable with 'baseline' and 'future' as levels) as explanatory variable.

Nisqually in our case study that negative allele counts and frequencies are predicted for one of the minor alleles in the RDA prediction step, but that the GAM step predicts the biologically acceptable frequency of 0.027. Function `AlleleShift::freq.ggplot` (Fig. 3)

**Table 2  Output from function `freq.pred` for the changed climate for four populations and 2 alleles (A and B).** This is a subset of the results with all populations and all alleles for the example data sets in *AlleleShift* for *Populus trichocarpa*.

| Population | Puyallup | Tahoe | Skagit | Nisqually |
|---|---|---|---|---|
| N | 372 | 24 | 326 | 28 |
| Locus | | X01_10838495 | | |
| Allele.freq | 0.172 | 0.625 | 0.236 | 0.107 |
| A | 64 | 15 | 77 | 3 |
| B | 308 | 9 | 249 | 25 |
| Ap | 144.376 | 23.471 | 75.381 | 15.165 |
| Bp | 227.624 | 0.529 | 250.619 | 12.835 |
| N.e1 | 372 | 24 | 326 | 28 |
| Freq.e1 | 0.388 | 0.978 | 0.231 | 0.542 |
| Freq.e2 | 0.425 | 0.959 | 0.192 | 0.633 |
| LCL | 0.306 | 0.794 | 0.175 | 0.452 |
| UCL | 0.544 | 1.000 | 0.210 | 0.813 |
| increasing | TRUE | TRUE | FALSE | TRUE |
| Locus | | X01_16628872 | | |
| Allele.freq | 0.169 | 0.083 | 0.181 | 0.250 |
| A | 63 | 2 | 59 | 7 |
| B | 309 | 22 | 267 | 21 |
| Ap | 38.751 | 0.509 | 52.668 | −0.038 |
| Bp | 333.249 | 23.491 | 273.332 | 28.038 |
| N.e1 | 372 | 24 | 326 | 28 |
| Freq.e1 | 0.104 | 0.021 | 0.162 | −0.001 |
| Freq.e2 | 0.104 | 0.037 | 0.183 | 0.027 |
| LCL | 0.091 | 0.029 | 0.161 | 0.018 |
| UCL | 0.117 | 0.045 | 0.205 | 0.036 |
| increasing | FALSE | FALSE | TRUE | FALSE |

**Notes.**

N, number of alleles sampled from population; `Allele.freq`, frequency of the minor allele in the baseline climate calculated via `adegenet::make.freq`; A/B, count of the alleles in the baseline climate; Ap/Bp, predicted counts of the alleles in the changed climate; N.e1, predicted number of alleles; Freq.e1: predicted frequency of the minor allele via RDA; Freq.e2, predicted frequency of the minor allele via GAM; LCL/UCL: lower/ upper confidence limits; increasing, flag whether frequency is increasing.

enables a visual inspection of the power of the models to predict allele frequencies for the calibration data.

## Visualizations

*AlleleShift* generates various types of *ggplot2* (*Wickham, 2016*; version 3.3.2) graphics from the output of `AlleleShift::freq.pred`. These include dot (Fig. 4), pie or donut (Fig. 5), moon (Fig. 6) and waffle (Fig. 7) graphics, and smoothed regression surfaces (Fig. 8). As an intermediate step to generate various of these graphics, helper functions such as `waffle.baker` for `shift.waffle.ggplot` or `moon.waxer` for `shift.moon.ggplot` prepare data for the main graphing function (Table 1). With default settings, visualizations depict changes in allele frequencies for each allele in different panels, internally via

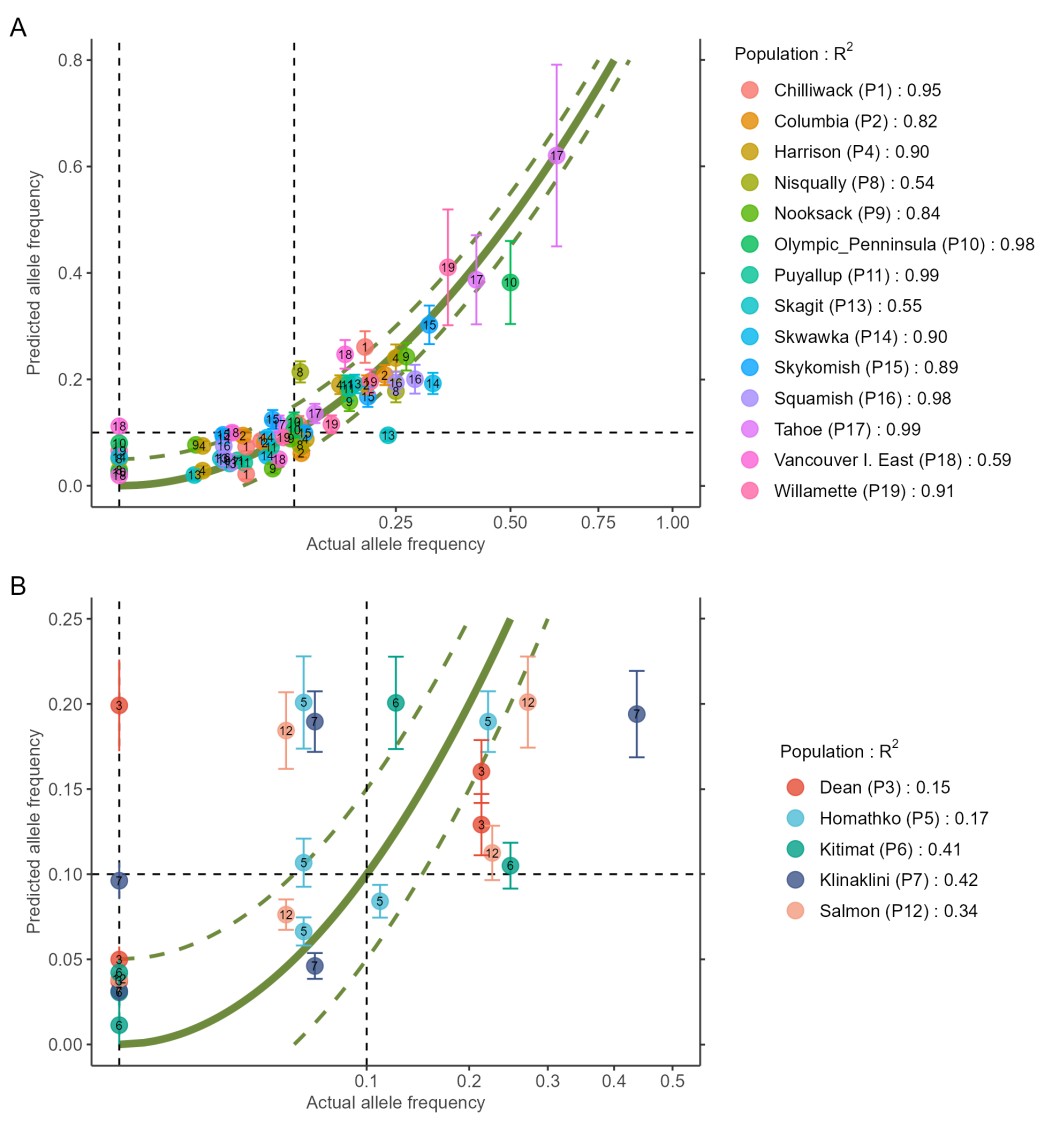

**Figure 3** **Plot of the actual frequency of the minor allele against the predicted frequency for the calibration data.** The 'olivegreen' reference lines indicate 1:1 (bold line), 1:1.05 and 1:0.95 (dashed lines) relationships. (A) Data for populations where a linear model explains more than 50% in allele frequencies. (B) Data for populations where a linear model explains less than 50% in allele frequencies.

`ggplot2::facet_grid`. Setting argument mean.change to TRUE, the graphics depict median or mean changes in allele frequencies.

Function shift.surf.ggplot plots populations in geographical space via their geographical coordinates (longitude and latitude in Fig. 8). The function fits a smoothed regression surface for allele frequencies via `vegan::ordisurf`. This output is then further processed internally within the function via `BiodiversityR::ordisurfgrid.long` (an overview of generating *ggplot2* ordination diagrams via *vegan* and *BiodiversityR* is given in *Kindt (2020b)*; these guidelines can be used to customize ordination graphs as shown in Figs. 2 and 8). Various options of fitting smoothed regression surfaces are available by providing

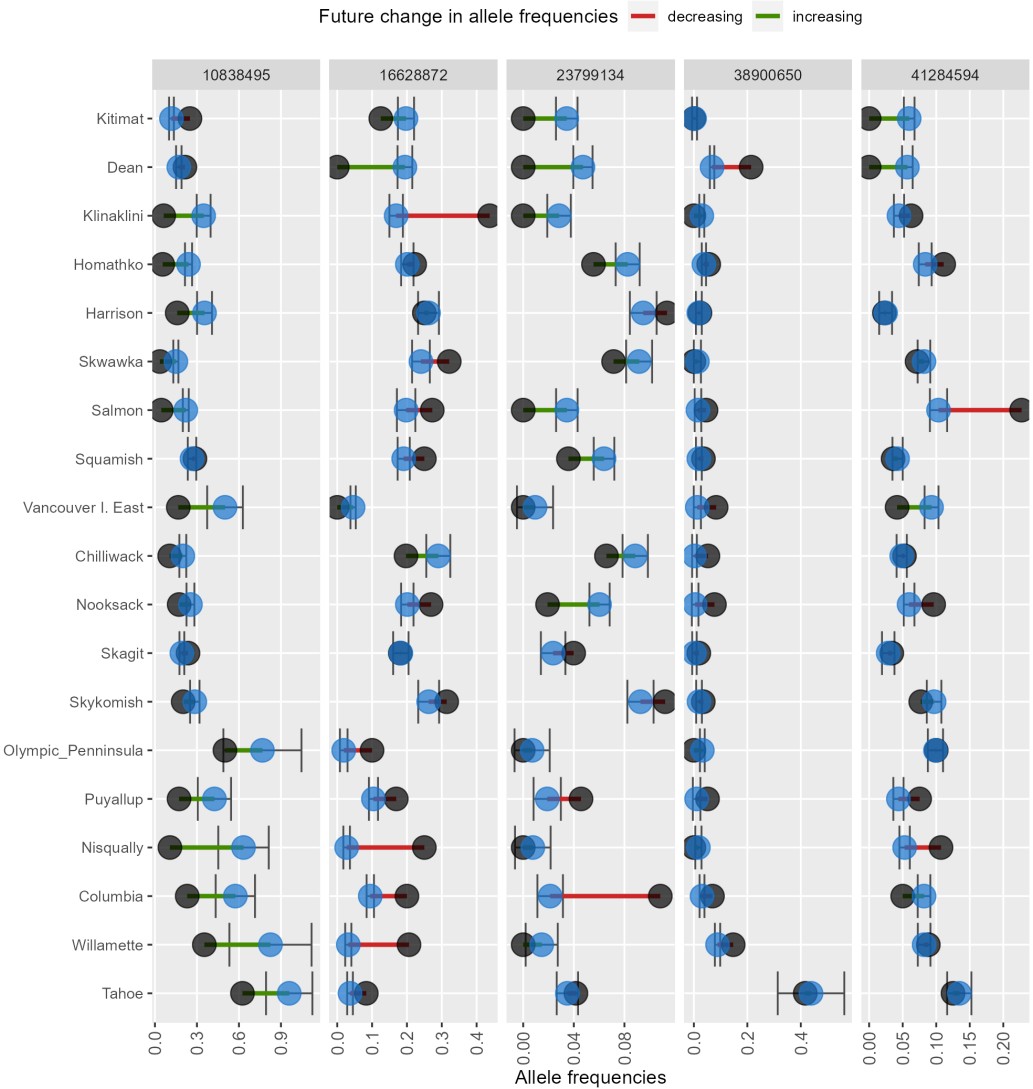

**Figure 4** **Depiction of changes in minor allele frequencies via AlleleShift::shift.dot.ggplot.** Black circles reflect baseline frequencies, blue circles indicate future frequencies and vertical lines indicate the confidence interval.

additional arguments to `shift.surf.ggplot`, such as the various `mgcv::smooth.terms` options of thin plates, Duchon splines, cubic regression splines, P-splines, Markov Random Fields, Gaussian process smooths, soap film smooths, splines on the sphere and random effects.

As graphics are generated with *ggplot2*, it is relatively easy to generate animated versions of visualizations with *gganimate* (*Pedersen & Robinson, 2020*; version 1.0.7). Scripts for generating animated versions for dot graphics, pie graphics and smoothed surfaces are provided with the documentation of the respective functions. Video S3–S5 are for a time series that interpolates bioclimatic data between baseline and future climate in steps of five years; these also extrapolate data far into the future (up to a million years, see the Results

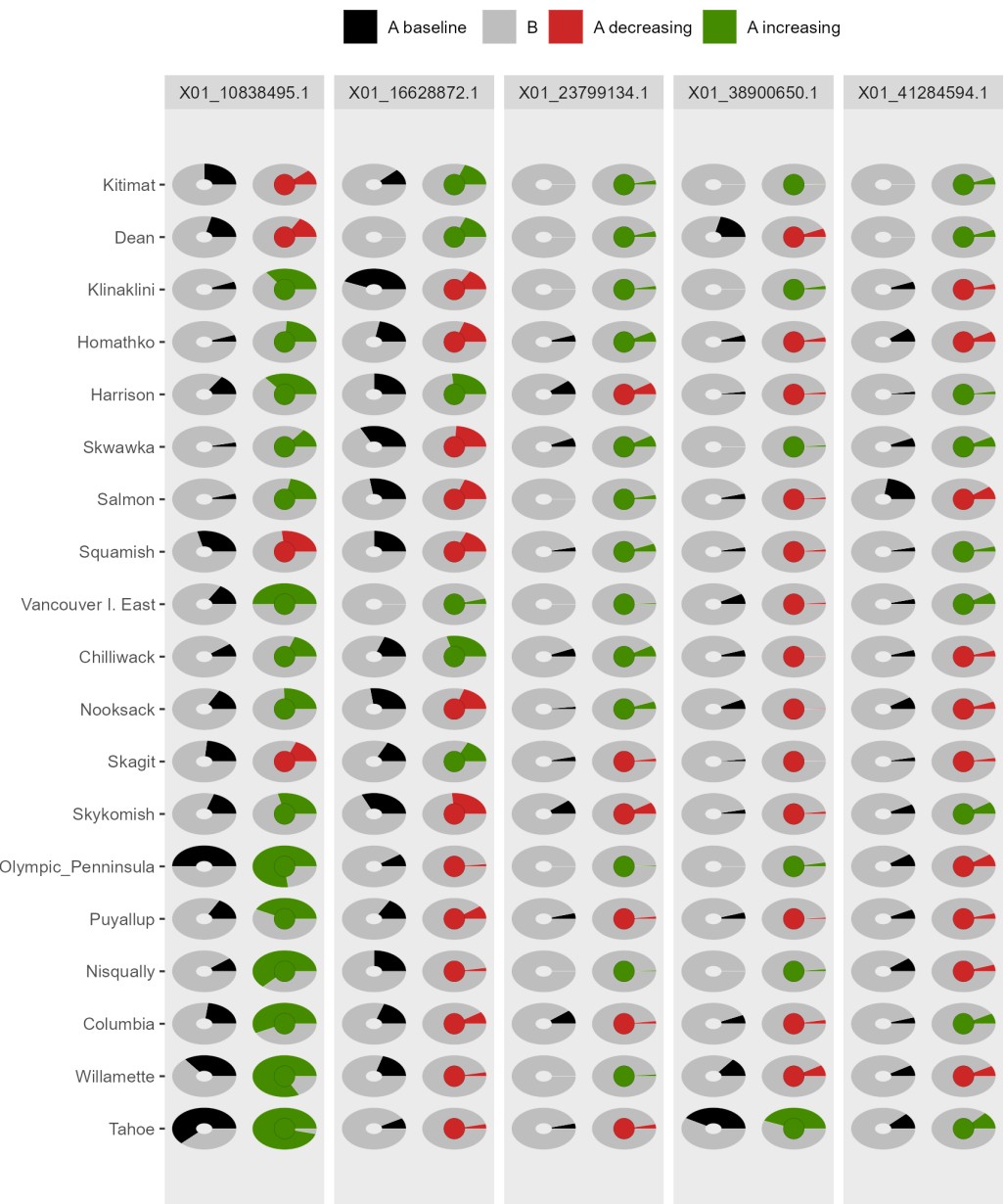

**Figure 5** **Depiction of changes in allele frequencies via AlleleShift::shift.pie.ggplot.** Columns on the left reflect baseline frequencies with frequency of the minor allele in black. Columns on the right reflect future frequencies, with colour of the arc and colour of the central circle reflecting frequencies and trends (red = decreasing, green = increasing) of the minor allele.

and Discussion for my reason to do this). Other than time series, animated visualizations could also be generated for different global circulation models (GCMs) or scenarios such as various shared socio-economic pathways (SSPs) developed in the context of the sixth assessment report of the Intergovernmental Panel on Climate Change.

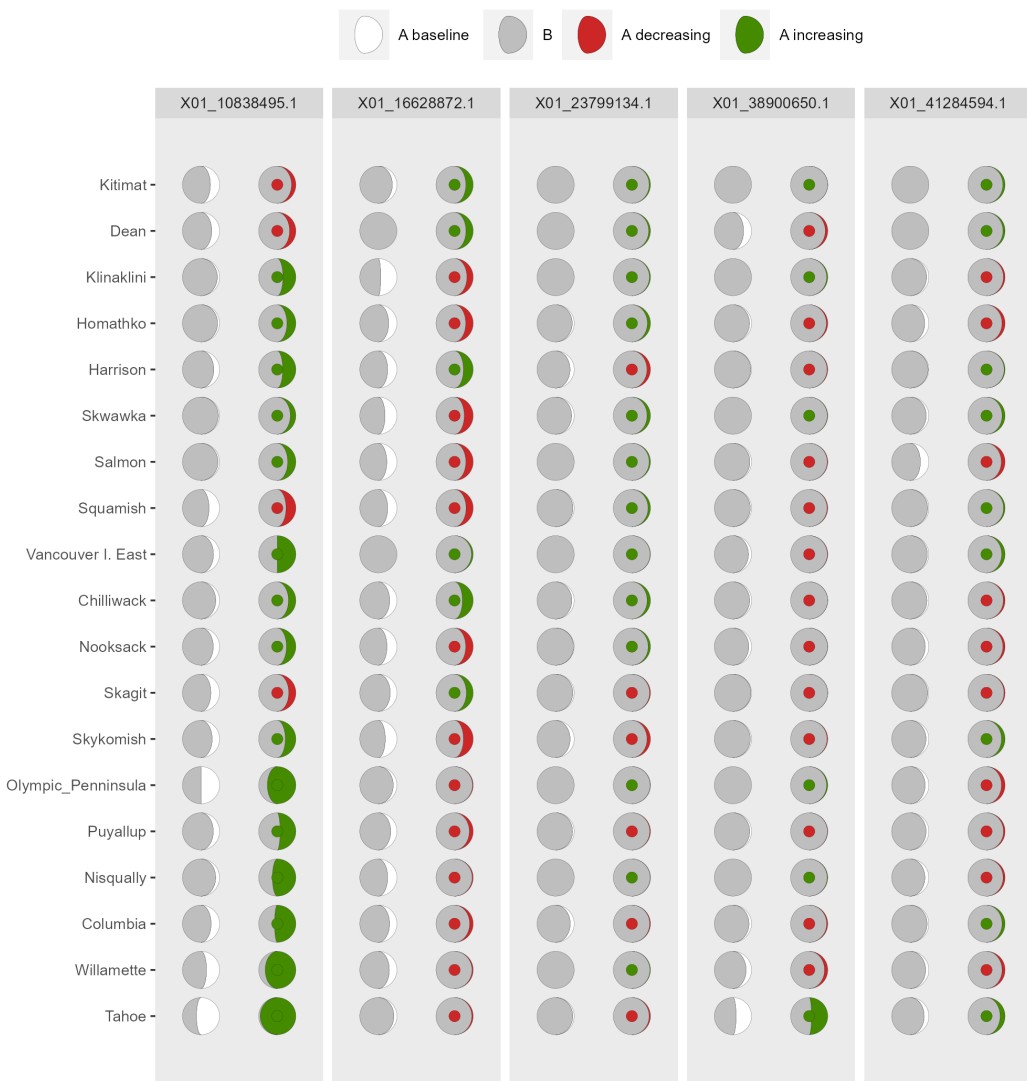

**Figure 6** **Depiction of changes in allele frequencies via AlleleShift::shift.moon.ggplot.** Columns on the left reflect baseline frequencies with frequency of the minor allele in white. Columns on the right reflect future frequencies, with colour of the waxing moon and colour of the central circle reflecting frequencies and trends (red = decreasing, green = increasing) of the minor allele.

## RESULTS AND DISCUSSION

*AlleleShift* predicts shifts in allele frequencies via RDA and GAM, an alternative pathway that maintains Euclidean distances among populations and individuals (Article S1). It also avoids making negative frequency predictions as what may occur with the protocol recently devised by *Blumstein et al. (2020)*. My methodology however faces the same limitations of data requirements as discussed by *Blumstein et al. (2020)* for their protocol in terms of the initial identification of responsive markers. Key assumptions of the *Blumstein et al. (2020)* protocol apply also to my approach, including that allelic effects are independent

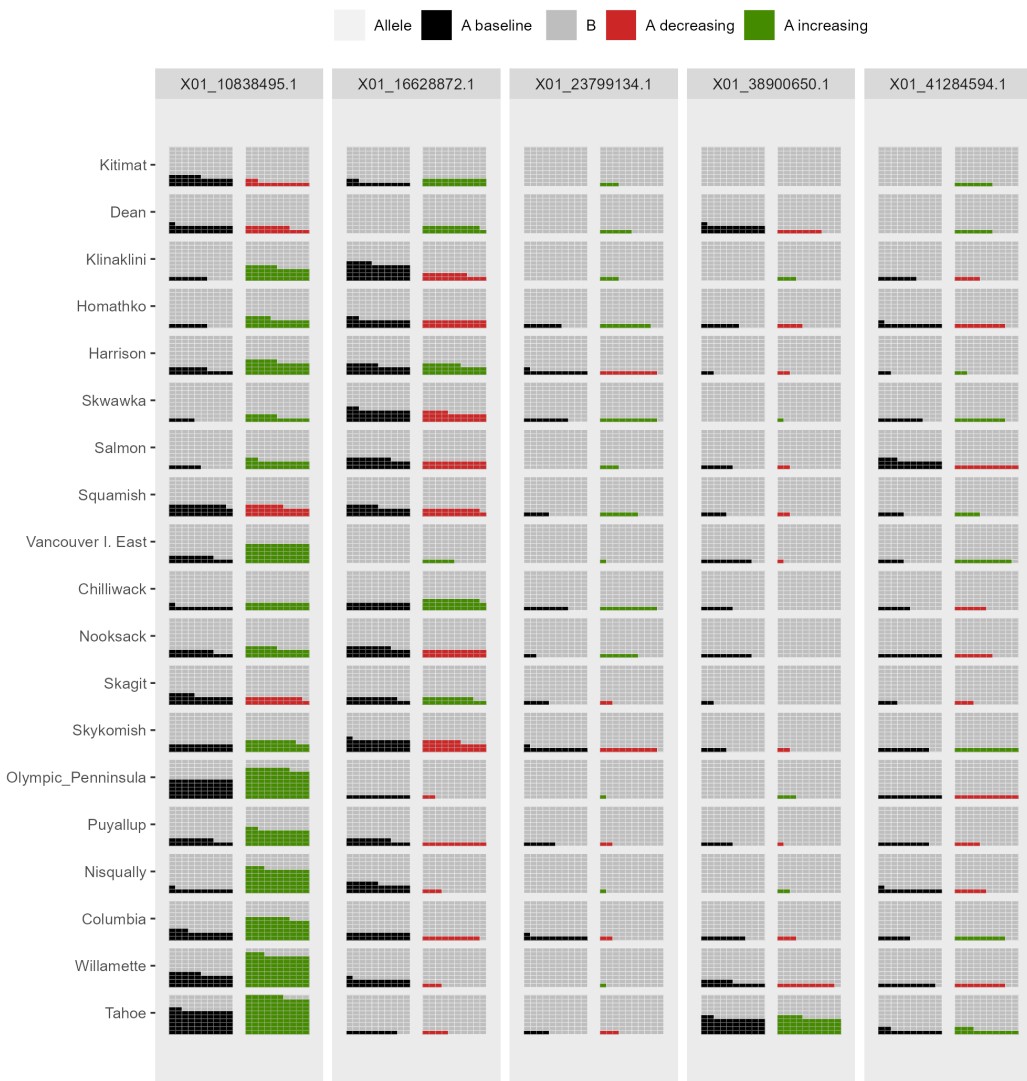

**Figure 7** **Depiction of changes in allele frequencies via AlleleShift::shift.waffle.ggplot.** Each 'waffle' has 100 'cells'. Columns on the left reflect baseline frequencies with frequency of the minor allele in black. Columns on the right reflect future frequencies of the minor allele, with colour indicating trends (red = decreasing, green = increasing).

and additive with no epistasis or dominance (see also the discussion on epistasis, structural genomic variation and epigenetics by *Stange, Barrett & Hendry (2020)*.

I recommend reducing both the explanatory variables and the response variables to subsets of data with a maximum Variance Inflation Factor (VIF) of 20 or less (*Ter Braak & Smilauer, 2002*), for alleles to obtain better estimates of changes in their frequencies. For the allele counts as response variables, if VIF would be larger than 20, I would use function VIF.subset in an iterative procedure whereby earlier selected subsets of alleles are excluded, to generate different subsets of alleles (but keeping variables for both A and B allele counts in each final subset). For example, with 20 alleles X01A, X01B, X02A, X02B,

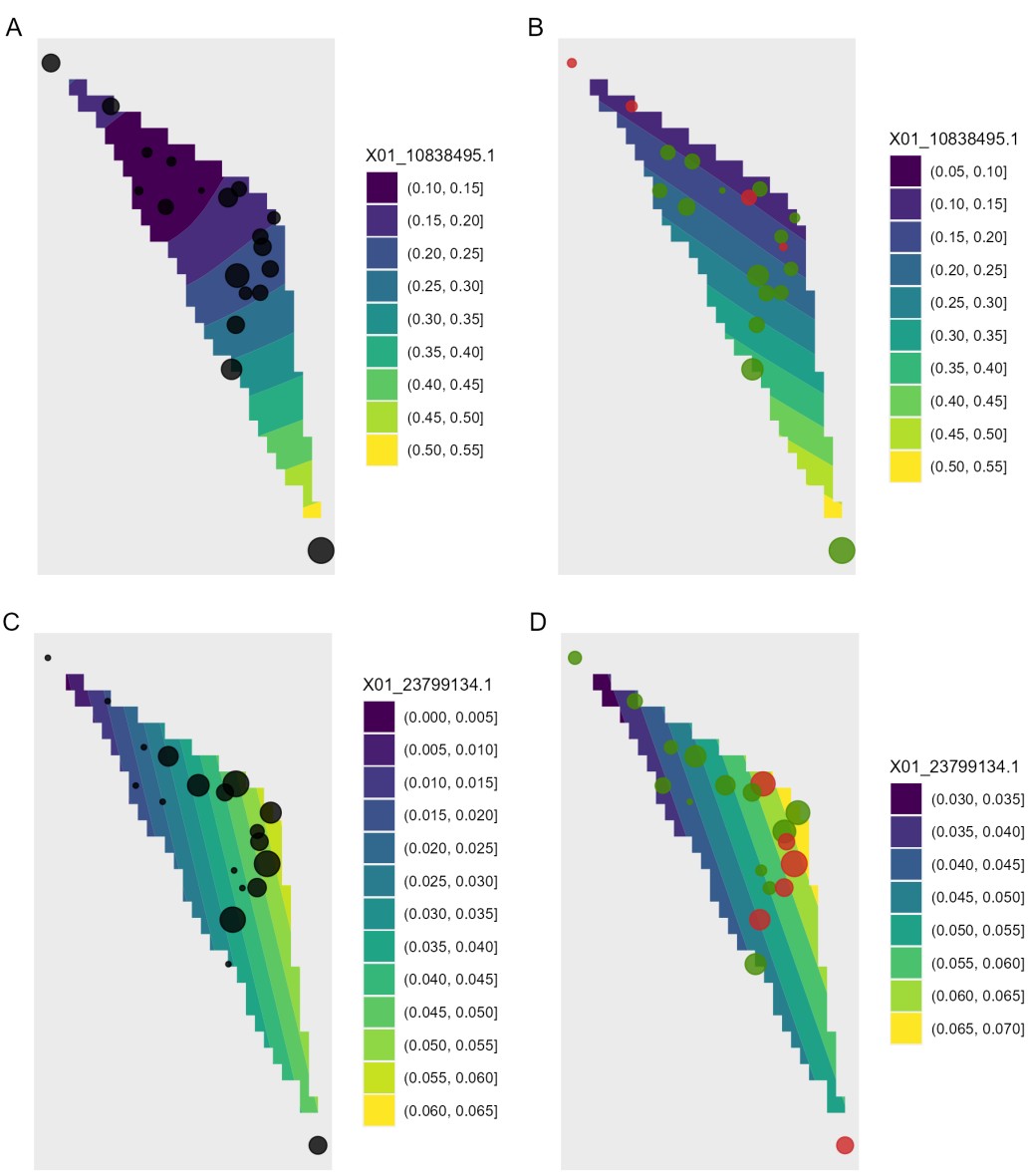

**Figure 8** Depiction of changes in allele frequencies in geographical space via AlleleShift::shift.surf.ggplot. (A) Frequencies in the baseline climate for the minor allele for locus 10838495. (B) Frequencies in a future climate for the same allele as in (A). (C) Frequencies in the baseline climate for the minor allele for locus 23799134. (D) Frequencies in a future climate for the same allele as in (D). Sizes of circles reflect the frequencies for the populations. Colours for the future frequencies (B, D) indicate trends (red = decreasing, green = increasing).

..., X10A, X10B, the two subsets could be {X01A, X01B, ..., X05A, X05B} and {X06A, X06B, ..., X10A, X10B}. With the various subsets, shifts in allele frequencies can then be predicted, and finally predictions with all subsets can be combined.

When predictions are made by *AlleleShift* into the future, and especially into novel climatic conditions, it is warranted to consider the transferability of the calibrated models and ideally to provide 'transferability metrics' that quantify prediction uncertainty (*Yates*

*et al., 2018*). The function of `environmental.novel` identifies populations that have novel climatic conditions (values outside the baseline range) for at least one of the explanatory variables for the changed climate. This function also calculates the probabilities of encountering the novel conditions from the mean and standard deviation of the baseline ranges and then returns the lowest of these probabilities as an indicator of the degree of novelty of the environmental conditions.

For the animated graphics, as an extreme example of the caveats of the methodology, I made projections one million years into the future. These projections followed earlier trends of the 21st century and resulted in allele frequencies becoming fixed at 0 or 1, which is a biologically possible scenario. At the same time, however, these simulations clearly illustrated that the methodology through RDA is correlative as in correlative/phenomenological species distribution models. In fact, the linear extrapolation of climate variables resulted in an environmental data set where the mean annual temperature was above 30,000 degrees, which obviously is a biologically irrelevant scenario. Thus, the method presented here should be used cautiously in novel climates especially as predictions will be available (the model will not crash or give an error warning), and should thus be used even more cautiously where differences between baseline and future climates are large.

As an approach to obtain a better handle on the transferability of *AlleleShift* predictions into future climates, I recommend to also estimate the habitat suitability via species distribution models (SDMs). Where habitat suitability is below a certain threshold, for instance a threshold where the sum of sensitivity and specificity is maximized (*Liu, White & Newell, 2013*), results of predictions of *AlleleShift* could be annotated of having lower transferability. Similarly, where evaluation strips as proposed by *Elith et al. (2005)* suggest that it is less likely that the habitat is suitable within the changed range of environmental conditions, additional information could be provided on a lower transferability. Well-documented methods of utilizing SDMs to predict shifts in species habitat suitability are available in the literature, including recent examples that use the ensemble suitability modelling framework available in *BiodiversityR* (e.g., *de Sousa et al., 2019*; *Fremout et al., 2020*; *Kindt, 2018*; *Ranjitkar et al., 2014*). For organisms such as trees, correlative SDMs remain the best available method of predicting future species suitability, whereas the limitations of these methods may not be as great as has been suggested (*Booth, 2016*; *Booth, 2018*). What is also attractive about SDMs is that a wider set of presence observations are likely to be available than those populations that have been studied genetically. Presence data are available from open-source databases such as GBIF or the Botanical Information and Ecology Network (*Enquist et al., 2016*). Further to the collation of a larger set of presence observations, application of SDMs should be straightforward using the same (bio)climatic data sets as applied in *AlleleShift*. The approach of combining the results of *AlleleShift* with SDM is somewhat similar to the method applied by *Aguirre-Liguori et al. (2019)* to develop species distribution models for alleles. In my proposal, however, the predictions of allele frequencies and SDM are done independently, and ideally with an expanded point presence data set for SDM.

A straightforward and practical expansion of the methodology I have proposed is to tree seed sourcing programmes (*Broadhurst et al., 2008*), possibly for developing schemes of human-assisted geneflow *sensu Aitken & Whitlock (2013)*. This is important for ensuring the matching of planting materials to the conditions of planting sites (*Cernansky, 2018*; *Roshetko et al., 2018*; *Kettle et al., 2020*). For specific planting sites and planting times (considering the perennial nature of trees, climate change during the production cycle should be considered) of interest, the prediction methods can readily provide the predicted allele frequencies needed for adaptation. Theoretically, based on the similarity between predicted allele frequencies and those of available source populations, the best matching source can then be selected. Similar approaches to devise transfer and conservation schemes in the face of climate change can be employed for other organisms than plants (*Fitzpatrick & Edelsparre, 2018*; *Rochat, Selmoni & Joost, 2021*).

## CONCLUSIONS

The R package *AlleleShift* provides a set of functions that allow the prediction of allele frequencies from baseline, future and past (bio)climatic explanatory variables via redundancy analysis (RDA) and generalized additive models (GAM). Various visualizations are provided via *ggplot2* and its extension packages such as *ggforce* and *gganimate*. At the time of submission of this manuscript, no package was available for this set of tools. As with any other methodology that projects into the future, it is important to reflect on transferability to novel climates.

## ACKNOWLEDGEMENTS

The author thanks Ian Dawson (CIFOR-ICRAF) for reviewing the article prior to submission and three anonymous reviewers who provided excellent suggestions to improve the manuscript. He also thanks additional colleagues from CIFOR-ICRAF and the University of Copenhagen for useful discussions on the applications of this package, including Lars Graudal, Prasad Hendre, Ramni Jamnadass and Jens-Peter B. Lillesø.

### Funding

The CGIAR Research Program on Forests, Trees, and Agroforestry (supported by the CGIAR Trust Fund) and the Provision of Adequate Tree Seed Portfolios project (supported by Norway's International Climate and Forest Initiative through the Royal Norwegian Embassy in Ethiopia) supported Roeland Kindt's time on this project. The funders had no role in study design, data collection and analysis, decision to publish, or preparation of the manuscript.

### Grant Disclosures

The following grant information was disclosed by the author:
CGIAR Trust Fund.

Norway's International Climate and Forest Initiative through the Royal Norwegian Embassy in Ethiopia.

## Competing Interests

The author declares there are no competing interests.

## Author Contributions

- Roeland Kindt conceived and designed the experiments, performed the experiments, analyzed the data, prepared figures and/or tables, authored or reviewed drafts of the paper, and approved the final draft.

## Data Availability

Code is available at CRAN: https://cran.r-project.org/package=AlleleShift and GitHub: https://github.com/RoelandKindt/AlleleShift

Data is available at GitHub: https://github.com/RoelandKindt/AlleleShift/tree/master/data.

Example scripts are available at GitHub: https://github.com/RoelandKindt/AlleleShift/blob/master/README.md.

## Supplemental Information

Supplemental information for this article can be found online at http://dx.doi.org/10.7717/peerj.11534#supplemental-information.

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
