# Peer review of "AlleleShift: an R package to predict and visualize population-level changes in allele frequencies in response to climate change"

_PeerJ, doi:10.7717/peerj.11534_

## Round 0.1 · original submission · Major Revisions

Dear Dr. Kindt:

Thanks for submitting your manuscript to PeerJ. I have now received three independent reviews of your work, and as you will see, the reviewers raised some concerns about the research. Despite this, these reviewers are rather optimistic about your work and the potential impact it will have on research studying population biology and informatics approaches for determining allelic frequencies. Thus, I encourage you to revise your manuscript, accordingly, taking into account all of the concerns raised by all three reviewers.

Please remove lengthy discussions comparing your approach with other protocols and just focus on the merits of your package. Other approaches, while needing to be cited, do not need to be overly referred to. Also, there is concern regarding the reliability of calibrating allele frequencies derived from an RDA using a GAM. The results from this procedure look biologically meaningful, as they avoid the negative values from the RDA outputs, and the values of both RDA and GAM are very similar to each other. Perhaps this could be better demonstrated with virtual datasets with known changes in the values of allele frequencies in changing climates (or using at least using one additional dataset to show that these predicted frequencies will be always similar between RDA and GAM).

While the concerns of the reviewers are relatively minor, this is a major revision to ensure that the original reviewers have a chance to evaluate your responses to their concerns. There are many minor problems pointed out by the reviewers, and you will need to address all of these and expect a thorough review of your revised manuscript by these same reviewers.

I look forward to seeing your revision, and thanks again for submitting your work to PeerJ.

Good luck with your revision,

-joe

Reviewer 1 ·

Basic reporting

The paper provides a very useful package for application by researchers in a hot development area. The output files look great.

Experimental design

no comment

Validity of the findings

no comment

Additional comments

Here are some suggestions for consideration:
22-23 alleles unrelated to adaptive traits may also change under migration so this needs some rewording.
33-36 This is critical but a little dense to follow. Given that RDA is used it might not be apparent to a reader why negative values are avoided.
103-105 Isn’t this a circular argument though? A linear analysis is more suitable for a linear pattern. I’d reword this.
106-108 There is of course plenty of evidence in the literature that clinal patterns in alleles along climate gradients can be non linear as well as linear. Might be worth a comment and I suspect that researchers will end up testing both approaches.
148-149 This could be restrictive given that SNP approaches typically do lead to some missing data. Bioclimatic data can always be interpolated but perhaps some guidance might be provided for the extent to which missing genetic data might be tolerated.
249-256 Not sure of the value of this! We know that the approach is correlative so as an exercise I suspect that this is not particularly useful.
258-275 The obvious challenge here is to think about how much difference evolutionary adaptation through allele shifts could make to shifts in species distributions as predicted by SDMs as per other papers like Aguirre-Ligouri and Bush et al (Ecology Letters) though the latter is based on heritability estimates. I suspect the authors could more clearly introduce this context here.
277 -285. There are quite a few papers on this that pre-date Aitken and Whitlock and discuss the broader issue of approaches around provenancing (see for instance Broadhurst, Prober etc in Frontiers in Ecology and Evolution). And of course, the challenge and application here is much wider than just considering plants, see recent Hoffmann et al paper in Evolutionary Applications.
General. There are of course some major limitations with the approach generally that go well beyond what is mentioned here. We don’t really know whether the correlative approach will tell us much about future adaptation in the absence of much validation. There have been critiques such as Fitzpatrick et al 2018 Science which should probably be mentioned somewhere.

Reviewer 2 ·

Basic reporting

This is a very nice piece reporting a new R package that allows modeling allele frequencies as a function of environmental or explanatory variables and, assuming adaptation or dispersal under conservative geographical patterns, to project the future expected allele frequencies under climate change. The paper is in general well written and I’m sure that a new nice and friendly user R package may trigger further applications of this area and highlight the need for more studies coupling genetic variation and ecological scenarios for climate change.

Experimental design

it is not an experimental paper

Validity of the findings

the script will be quite useful indeed

Additional comments

This is a very nice piece reporting a new R package that allows modeling allele frequencies as a function of environmental or explanatory variables and, assuming adaptation or dispersal under conservative geographical patterns, to project the future expected allele frequencies under climate change. The paper is in general well written and I’m sure that a new nice and friendly user R package may trigger further applications of this area and highlight the need for more studies coupling genetic variation and ecological scenarios for climate change.

The package is well designed, with nice functions and great visualization tools, and, most important, well compatible with distinct dataset formats commonly used in population genetics packages. Also, it is based on and actually uses several functions from Vegan, which is the most important package (in my opinion) for data analysis (especially multivariate analysis) in Ecology.

Thus, I have just to minor concerns that hopefully may help authors improve a bit the manuscript.

First, although I understand the worry of the author about the (slight) changes in respect to the other protocol proposed by Blumstein et al., it is weird to go into such details in a paper describing a package. Perhaps a few lines in the introduction and discussion are enough, and my impression is that author is perhaps “over-reacting” to previous comments or suggestions. If so, I suggest deleting most of the final part of the introduction and Table 1. So, the paper would be more focused on the package. Perhaps the author would like to compare the two protocols or approaches, but I think this would be a completely different paper, analyzing empirical data and, even better, simulated data (and if author decides to go with this idea here, the R-package would be a supplement or something like this). Anyway, regardless of the strategy, presenting the package and comparing approaches are two different ideias.

Another minor issue, in the “methods”, “data analysis” section, I would like to see a bit more detail on projections. There is a lot of emphasis on calibration, and I understand this, but I barely see something on projection. Also, I’m not sure I followed why goes with an RDA and the use of GAM, it is not clear (perhaps my fault).

Reviewer 3 ·

Basic reporting

Language: There are a few minor corrections to the language that do not change the meaning of the text:

Line 26-27 - “AlleleShift of predicting changes in allele frequencies at populations’ locations.”
- Perhaps: “at the population level or for each location”

Line 40 - “These include ‘dot plot’ graphics”
- These outputs include…
- Also, in several instances, you use "This" and "These" at the begging of a sentence relating to the subject of the sentence before. Although English is not my native language, perhaps you should always write down what "this" and "these" are related to in each sentence as in the suggestion above.

Line 204 “Various options of fitting smoothed regression surfaces are available through providing”
- Change to: by providing

Line 225 – “…negative frequency predictions as can occur with the CCA approach”
- Maybe rephrase to: as what/it may occur with the CCA approach?


Figure 1 - Add what variables are the acronyms in the figure.

You have done a good job of providing updated references and figures, tables and raw data that are meeting the standards of the journal.

Experimental design

Line 104-105 – Perhaps add a caveat here or in the discussion that not all changes are expected to be linear.

Lines 130-131 – There are also the CHELSA and CLIMOND databases with current and future predictions of climatologies.

Line 148-149 – “1). I further advise to remove any individuals with partially missing
genetic or (bio)climatic data prior to the analysis.”
- Would you advise to estimate missing allele states for these individuals using the LEA package or something else?

Lines 156-157 – “What is important to check in the ordination graphs is whether populations shift in a similar fashion”
- And what if they don’t change in a similar fashion? Will the method still provide reliable results? Perhaps add a sentence to help the user.

Lines 226-228 – “My methodology however faces the same limitations of data requirements as discussed by Blumstein et al. (2020) for their protocol in terms of the initial identification of responsive markers.”
- Shouldn't this limitation be mentioned in the methods? If I understood you correctly, you have to remove neutral loci, right? Maybe also advise which methodologies one could use to solve this caveat prior to applying AlleleShift

Lines 233-234 – “I recommend reducing both the explanatory variables and the response variables to subsets of data with a maximum Variance Inflation Factor (VIF) of 20 or less”
- Here you should provide a reference. I have seen values of 5 or 10 for the VIF procedure in the literature.

Lines 236-237 – “if VIF would be larger than 20, I would use function VIF.subset in an iterative procedure whereby earlier selected subsets of alleles are excluded”
- I understood that VIF.subset would only remove environmental variables, right? Perhaps include/expand this explanation in the methods section instead of in the results.

Lines 244-245 – “it is warranted to consider the transferability of the calibrated models and ideally to provide ‘transferability metrics’ that quantify prediction uncertainty (Yates et al., 2018).”
- Could this be provided in the AlleleShift documentation/scripts as well with the dataset that you are using?

Lines 258-261 – “As an approach to obtain a better handle on the transferability of AlleleShift predictions into future climates, I recommend to also estimate the suitability (and ideally the transferability, for instance via evaluation strips as proposed by Elith et al., 2005) of target species via species distribution models (SDMs).”

- I am not sure that I understand how to use the SDM results to handle the transferability of AlleleShift predictions. How would you actually do it? Or would you just compare/visualize both outputs across the geographical space? It would be nice to make this clearer for the user. For example, when would you not calculate the AlleleShift predictions depending on the evaluation strips that you mentioned? Finally, could you do the same for the dataset used in your manuscript to show that the data you are using is transferable?

Validity of the findings

Lines 167-168 – "This procedure ensures that predictions are within the realistic interval for frequencies between 0 to 1.”

I understand the need for having allele frequencies in the interval between 0 and 1. I can see that the RDA and GAM results with your dataset are very similar. But as a speculation, is it possible to estimate known changes in allele frequencies with a virtual dataset just to check whether AlleleShift is providing the “true” answer? Or at least test these results with an additional dataset to see if the results are always similar between RDA and GAM?

Additional comments

I believe that your manuscript is interesting, the methodology is useful, and that there are mostly small corrections to be made. Here some general comments:

Lines 23- - “with adaptive traits are expected to change in future climates through local adaption and migration”.
- As you mention migration here, perhaps indicate in the discussion that migration is not being considered in your methodology (or indicate whether migration could mask the effects of predicted changes in allele frequencies not related to adaptation).

Abstract – Methods: I think you should start with the input data, for example, current climate data, predicted climates, SNPs/genomic data.

Lines 31-32 - “As such, the procedure is fundamentally different to an alternative approach
recently proposed to predict changes in allele frequencies from canonical correspondence analysis (CCA).”
- Here it is missing a sentence with some information on why this procedure is better than CCA (as you indicate in the introduction/methodology).

Line 38 - Perhaps: “AlleleShift provides data sets with predicted frequencies, several visualization methods..”

Line 55 – “There is clear evidence of climate change”
- When? Perhaps add: There is clear evidence of anthropogenically induced climate change.

Line 56 – “many countries are developing National Adaptation Plans (NAPs)”
- Here a small explanation is needed at the end of this sentence, what is a NAP?

Line 63 – I am missing a sentence or two providing a link between climate change and changes in the genome/allele frequencies.

Line 70 – Here I believe that you should first introduce the method in a separated sentence and then compare it with Blumstein et al. (2020). What is AlleleShift? Is it already online and being widely used?

Line 261-262 – “utilizing SDMs to predict shifts in species habitat”
- I would not say for species habitat but for species habitat suitability.

Table: “Fit superellipses and conduct RDA to check for niche differentiation between baseline and future positions of climates”
- Niche differentiation sounds like something different, perhaps check for changes in climate conditions across locations between baseline and future climates?

---

## Round 0.2 · accepted · Accept

Dear Dr. Kindt:

Thanks for revising your manuscript based on the concerns raised by the reviewers. I now believe that your manuscript is suitable for publication. Congratulations! I look forward to seeing this work in print, and I anticipate it being an important resource for groups studying population biology and informatics approaches for determining allelic frequencies. Thanks again for choosing PeerJ to publish such important work.

Best,

-joe

Reviewer 1 ·

Basic reporting

I think this is fine.

Experimental design

My earlier comments have been addressed.

Validity of the findings

Earlier comments have been addressed.

Additional comments

Thanks for the revisions, happy to see this published.

Reviewer 2 ·

Basic reporting

.

Experimental design

.

Validity of the findings

.

Additional comments

.

Reviewer 3 ·

Basic reporting

The manuscript has improved a lot in this new version. The language is clearer and the text is nice to read. References are updated and more than enough to provide the basis for research using the package described. All figures and tables are useful and well designed.

Experimental design

The methodology is well described and all concerns that I had before regarding the comparison of RDA and GAM allele frequencies in a simulated or additional dataset were satisfactorily answered. Now the goals of the manuscript appear clear to me, with a better focus on explaining the new method instead of exhaustively comparing it with others. The supplementary material and the material on GitHub allow direct access to the raw data and all steps needed for the analyses. It was clear to me how this research fills the gap on the topic of predicting allele frequencies into distinct scenarios of environmental change.

Validity of the findings

I believe that the new package and methodology will be very welcomed and the author clearly indicated the impact and novelty of it in the manuscript. The results with the test datasets are very interesting, the underlining data was provided, and it looks easily reproducible, given that every step is documented on the package webpage provided. The results were well linked with the conclusions and with the suggestions on how/when to use or not use this method. The reviewing processes really made a difference for the improvement of this manuscript.

Additional comments

The manuscript has improved a lot in this new version. I am glad to see how easily I could read it this time, almost without any interruption, understanding every step. You did a great job of incorporating the many suggestions of the reviewers. As a potential user, I could also follow well the online material on GitHub and it really does look easy to apply to any other datasets. I do not even have any specific comments on language and structure this time, so these are all my comments.